# Genetic Dissection of CRISPR-Cas9 Mediated Inheritance of Independently Targeted Alleles in Tobacco *α-1,3-Fucosyltransferase 1* and *β-1,2-Xylosyltransferase 1* Loci

**DOI:** 10.3390/ijms23052450

**Published:** 2022-02-23

**Authors:** Hayoung Song, Ju-Young Ahn, Fanzhuang Yan, Yidong Ran, Okjae Koo, Geung-Joo Lee

**Affiliations:** 1Department of Horticulture, Chungnam National University, Daejeon 34134, Korea; hysong@cnu.ac.kr; 2Department of Smart Agriculture Systems, Chungnam National University, Daejeon 34134, Korea; wnduds357@naver.com (J.-Y.A.); yanfanzhuang@126.com (F.Y.); 3Genovo Biotechnology Co., Ltd., Tianjin 301700, China; yidongran@genovo.org; 4Toolgen, Inc., Seoul 08594, Korea

**Keywords:** allele heredity, Cas9-free, multi-locus mutation, *Nicotiana benthamiana*, segregation ratio

## Abstract

We determined the specificity of mutations induced by the CRISPR-Cas9 gene-editing system in tobacco (*Nicotiana benthamiana*) alleles and subsequent genetic stability. For this, we prepared 248 mutant plants using an *Agrobacterium*-delivered CRISPR-Cas9 system targeting *α-1,3-fucosyltransferase 1* (*FucT1*) and *β-1,2-xylosyltransferase*
*1* (*XylT1*) genes, for which the mutation rates were 22.5% and 25%, respectively, with 20.5% for both loci. Individuals with wild-type (WT) alleles at the *NbFucT1* locus in T_0_ were further segregated into chimeric progeny (37–54%) in the next generation, whereas homozygous T_0_ mutants tended to produce more (~70%) homozygotes than other bi-allelic and chimeric progenies in the T_1_ generation. Approximately 81.8% and 77.4% of the homozygous and bi-allelic mutations in T_0_ generation, respectively, were stably inherited in the next generation, and approximately 50% of the Cas9-free mutants were segregated in T_2_ generation. One homozygous mutant (Ta 161-1) with a +1 bp insertion in *NbFucT1* and a −4 bp deletion in *NbXylT1* was found to produce T_2_ progenies with the same alleles, indicating no activity of the integrated Cas9 irrespective of the insertion or deletion type. Our results provide empirical evidence regarding the genetic inheritance of alleles at CRISPR-targeted loci in tobacco transformants and indicate the potential factors contributing to further mutagenesis.

## 1. Introduction

Breeding strategies are continually being modified in response to changing climates and for the development of crops with superior traits [1]. Although traditional breeding, based on crossing and selection, has made important contributions to the development of crops with improved characteristics, it has the disadvantage of requiring high investments in space, time, and finance [2]. Compared with traditional breeding, the introduction of molecular breeding based on marker-assisted selection has enhanced selection efficiency; however, this approach also has limitations in that it is difficult to identify useful selection markers and procedures that tend to be hampered by complex genetic variables such as polyploidy, heterozygosity, or self-incompatibility [2,3]. Plant transgenic breeding generally overcomes these major limitations by introducing beneficial exogenous genes or suppressing the expression of undesirable endogenous genes in crops and can accordingly play a pivotal role in crop improvement [4]. However, owing to safety issues regarding the potential effects of introduced foreign genes, transgenic crops are subject to strict regulations related to their use and tend not to be widely adopted [4]. An alternative to solving these problems is the recent active introduction and utilization of gene-editing technologies. Gene-editing tools include zinc finger nucleases, transcription activator-like effector nucleases, and the clustered regularly interspaced short palindromic repeats CRISPR-Cas9 system [5], which is one of the most recently adopted gene-editing technologies that has received the most attention over the past decade [6,7].

The CRISPR-Cas9 system consists of three elements: the Cas9 protein, gRNA, and target sites directly upstream of a protospacer adjacent motif (PAM; 5′-NGG-3′ for SpCas9 derived from *Streptococcus pyogenes*) [1]. gRNA consists of a trans-activating CRISPR RNA (tracrRNA) portion that forms a complex with Cas9 and a CRISPR RNA (crRNA) portion that can undergo complementary binding to a target sequence [5]. Cas9 nuclease recognizes a PAM sequence at the 3′ end of the target sequence and induces a double-strand break in the 3 bp portion upstream of the PAM. The damaged DNA is reassembled via nonhomologous end-joining (NHEJ) or homology-directed repair pathways [6]. In the case of NHEJ, an insertion or deletion is introduced in the gene-coding region to generate frameshift mutations or early stop codons, thereby generating gene-edited crop plants in which the function of the target gene is knocked out [8]. Compared with other gene-editing technologies, the CRISPR-Cas9 system has advantages such as the efficiency of target site design, potential for simultaneous multiple editing, and simplicity that does not require complex protein engineering [9]. Given these advantages, this technology has been used to modify a wide range of plants including rice, corn, wheat, soybean, barley, tomato, petunia, and lettuce [5,6,10].

To date, however, most research in this field has focused on the development of T_0_ or T_1_ generation transformants in which a target gene has been edited using the CRISPR-Cas9 system. To complete the development of stable transgenic plants, it is also essential to investigate how the modified gene is transmitted through subsequent generations and to assess the potential appearance of unanticipated mutations. Michno et al. [11] and Pan et al. [12] analyzed the transmission pattern of mutations induced by the CRISPR-Cas9 system to the next generation in soybean and tomato, respectively, whereas Feng et al. and Jiang et al. [13,14] reported the inheritance of mutant genes generated using Cas9/sgRNA in T_2_ generation *Arabidopsis* plants. The findings of these investigations can be summarized as follows: when a mutation that occurs in T_1_ is not transferred to T_2_, when a mutation that does not exist in T_1_ occurs in T_2_, when a mutation that occurs in T_2_ and T_3_ is transferred according to the Mendelian model or shows a non-Mendelian tendency, it shows a variety of separation patterns, indicating that these complex outcomes may arise from several potential mechanisms [11,14].

Tobacco (*Nicotiana benthamiana*) is one of the most widely used plant hosts in virological studies that have made important contributions to tissue culture procedures and the development of genetically engineered plants. Moreover, in recent years, tobacco has been evaluated as a potential biofactory for the production of biopharmaceuticals [15,16,17]. To realize this potential, there has been an expansion in the development of transgenic tobacco in which various genes have been targeted using the CRISPR-Cas9 system [18,19,20]. Currently, however, there has been insufficient evaluation of the specificity of mutations induced by the CRISPR-Cas9 system in tobacco, as well as the stability of progeny inheritance. Consequently, to ensure the development of transformant tobacco in which the acquired useful traits are stably maintained through multiple generations, it is necessary to provide information on the genetic pattern, efficiency, and specificity of CRISPR/Cas-induced gene mutations in different cases. Plants, including tobacco, produce two plant-specific N-glycans (β-1,2-xycoside and α-1,3-fucoside) [17]. Despite having many advantages as a recombinant protein production system, applications for massive production platforms are limited because plant-specific N-glycans may adversely affect protein activity and immunogenicity [17]. Therefore, for the development of plant materials capable of producing nontoxic N-glycan-type glycoproteins, two genes encoding the glycosyltransferase enzyme involved in the synthesis of plant-specific N-glycans, *α-1,3-fucosyltransferase* and *β-1,2-xylosyltransferase*, may be important targets for gene editing [19]. In this study, we produced transgenic *N. benthamiana* plants, in which we targeted the *α-1,3-fucosyltransferase 1* (*NbFucT1*) and *β-1,2-xylosyltransferase 1* (*NbXylT1*) genes to evaluate gene editing using the CRISPR-Cas9 system in tobacco and assessed how these modifications are inherited and segregated in subsequent generations. Furthermore, we analyzed the mutation transfer pattern in the next generation (primarily T_1_, and in some cases, T_2_).

## 2. Results

### 2.1. Generation of CRISPR/Cas9-Mediated Transgenic N. benthamiana

*N. benthamiana* transformants were produced using a CRISPR-Cas9 system targeting *XylT1* and *FucT1*. gRNAs were constructed to specifically target the first exon region (reverse) of *NbXylT1* (Niben101Scf04551) and *NbFucT1* (Niben101Scf01272), and the designed gRNAs were evaluated for potential off-target activity (Figure 1A, Appendix A). A single vector carrying gRNAs independently targeting *NbFucT1* and *NbXylT1* was constructed in the same plasmid (designated as pNbFUT1-NbXYLT1) and coexpressed with Cas9. Expression of the two gRNAs was driven by the *Arabidopsis* U6 promoter, whereas that of Cas9 was driven by the CaMV35S promoter (Figure 1B). In total, 248 plants transformed with *Agrobacterium*-delivered pNbFUT1-NbXYLT1 editing construct were regenerated (Figure 1C).

In these plants, the target region of *NbFucT1* or *NbXylT1* was amplified via PCR for Sanger sequencing, and an estimate of the efficiency of Cas9/sgRNA-mediated mutagenesis was obtained based on an analysis of the resulting sequencing chromatograms. We identified 56 and 62 plants with mutations in the *NbFucT1* and *NbXylT1* targets, respectively (Table 1). Moreover, we identified 51 plants in which mutations occurred simultaneously in both the target sequences. Specifically, we obtained mutation efficiencies of 22.5% and 25% for *NbFucT1* and *NbXylT1* targets, respectively, with an average mutation rate of 23.7%. The rate of simultaneous mutations in both targets was 20.5%, and the proportion of dual mutations within the mutant population was 76%.

### 2.2. Identification of CRISPR-Cas9-Induced Mutations at Target Sites in T_0_ Plants

To confirm the genotype of the Cas9/sgRNA-induced mutations, the sequence of mutations that occurred across the target site in 17 random representative transgenic lines was followed. Of the 17 independent transgenic lines analyzed, three lines (Ta75, Ta139, and Ta158) had mutations only at the *NbXylT1* target site, and one line (Ta77) had mutations at the *NbFucT1* target site alone. In the remaining 13 lines (Ta3, Ta12, Ta18, Ta27, Ta105, Ta106, Ta146, Ta161, Ta122, Ta117, Ta133, Ta237, and Ta248), mutations were detected in both target regions (Table 2, Figure 2). In all instances, mutations occurred as either an insertion or deletion of nucleotides upstream of the PAM sequence in a target site-specific manner without compromising the characteristics of Cas9/sgRNA catalyzed DNA cleavage and NHEJ DNA repair. The proportions of zygosity, according to the type of mutation occurring at the *NbFucT1* gene target site, were distributed as follows: wild type, 17%; homozygous mutation, 52.9%; and bi-allelic mutation, 29.4%. By contrast, the *NbXylT1* target region showed a 94% bi-allelic mutation and 5.8% wild type. With respect to the bi-allelic type of mutations identified, we observed that insertions occurred primarily as single-base additions, whereas in the case of deletions, excision of 1, 4, 9, and 11 bp was observed. Similarly, in the homozygous mutants, 1 bp additions were the predominant type of insertions, whereas deletions of more than 4 and 9 bp were detected. Accordingly, using the CRISPR-Cas9 system, insertions occurred predominantly at the single-base pair level, whereas deletions tended to be more extensive, ranging from 1 to 11 bp or more.

### 2.3. Inheritance of T_0_ Variant Target Mutations Evaluated in T1 Plants

To characterize the pattern of transmission of CRISPR-Cas9-mediated mutations to the T_1_ generation, we examined the genotypes of the two target regions in selected lineages (six independent lines) of T_0_ transgenic plants with confirmed mutagenesis. For each lineage, we performed an in-depth analysis of 8–11 T_1_ generation plants using targeted deep sequencing. In the case of four T_0_ lines (Ta105, Ta106, Ta146, and Ta161) that had a homozygous mutant genotype at the *NbFucT1* target site, the same homozygous genotype was detected in 70–90% or more of the T_1_ generation plants (Table 3). This supports the hypothesis that, in the case of homozygotes, the gene modification can be stably passed on to a significant proportion of the next generation. However, the remaining 10–30% of T_1_ generation plants were characterized by a bi-allelic or chimeric genotype, with additional mutations that were not detected in the previous generation. In the progeny of the Ta75 and Ta139 lines lacking a mutation in the *NbFucT1* target site, 54.5% and 37.5% were chimeric, respectively, and only 45.5% and 62.5% inherited the wild-type trait (Table 3).

T_0_ lines characterized by bi-allele-type mutations in the *NbXylT1* target region segregated into more diverse forms in the T_1_ generation. In the three lineages Ta75, Ta139, and Ta106, 36.4–37.5% of T_1_ generation plants received an unmodified bi-allele mutation, among which 37.5–50% were isolated as homozygotes (Table 4). In general, if the bi-allelic genotype is inherited according to Mendel’s laws, segregation of 1xx:2xy:1yy can be expected in the T_1_ generation. In the Ta75, Ta139, and Ta106 lines, we observed that the ratio of homozygous to bi-allelic plants in the T_1_ generation was close to 1:1, indicating that the CRISPR-Cas9-mediated bi-allelic mutation in tobacco has a Mendelian inheritance tendency. However, the segregation ratio between the isolated homozygotes was not limited to 1xx:1yy. The segregation ratios between the homozygotes of Ta75, Ta139, and Ta106 were 2xx:3yy, 3xx:0yy, and 3xx:1yy, respectively. Therefore, it is necessary to accumulate additional information by increasing the number of samples analyzed. In the remaining three lineages (Ta105, Ta146, and Ta161), bi-allelic mutations occurring in the ancestral *NbXylT1* target were also found to segregate into homozygous and bi-allelic mutations in the subsequent T_1_ generation. However, we found that T_1_ progenies segregated in several different ratios, including 1:2.5 (two homozygous–five bi-allelic plants), 1:5 (one homozygous–five bi-allelic plants), and 2:1 (six homozygous–three bi-allelic plants). Similar to the *NbFucT1* target site, we established that the *NbXylT1* target site was characterized by unexpected mutations in T_1_ plants. In most T_1_ lines, a chimeric genotype was detected at approximately 10%, whereas 30% of Ta146 line plants were characterized by this genotype (Table 4). We also identified T_1_ plants with mutations that were not derived from the mutant genotypes identified in the T_0_ generation. Among them, Ta75-16, Ta139-15, and Ta146-15 had a bi-allelic genotype, whereas Ta105-3 and Ta105-4 had a homozygous genotype.

Based on these observations, we established the probability level of developing T_1_ plants that stably inherited genotype T_0_ without additional unexpected mutations. The probability of occurrence of a plant in which the wild-type genotype at T_0_ was maintained at T_1_ was at an average of 53% (see Ta75 (45%) and Ta139 (62.5%) in Table 3), and the probability of occurrence of a plant with the homozygous mutation retained was at an average of 81% (see Ta105 (80%), Ta106 (87.5%), Ta146 (70%), and Ta161 (90%) in Table 3). In addition, when a bi-allelic type mutation occurred, the probability of maintaining or segregating as expected from T_1_ was calculated at an average of 77.4% (refer to the numbers of homozygous and bi-allelic mutations in each mutant line, except for the unexpected mutations in Table 4).

### 2.4. Inheritance and Stability of T_2_ Plants Derived from Homozygotes

We further investigated the inheritance and stability of mutations in the T_2_ generation progeny derived from the T_1_ generation Ta161-1 line, which had inherited mutations occurring in each of the *NbFucT1* and *NbXylT1* target sites in the homozygous form and retained Cas9. Analysis of the genomic region spanning the target site in 20 T_2_ generation lines based on deep sequencing revealed that in all 20 T_2_ lines investigated, the T_1_ genotype was unchanged, and we detected no additional unexpected mutations, which were notable features of inheritance from the T_0_ to T_1_ generation (Table 5). Furthermore, based on an examination of the inheritance of Cas9 via gDNA PCR analysis, we established that the T_1_ generation plant Ta161-1, which retained Cas9, passed this on to 50% of the assessed T_2_ generation progeny.

Thus, these observations indicate that if T_1_ generation plants have a homozygous mutant genotype, the target mutation can be stably transmitted to the next generation regardless of the presence of Cas9.

## 3. Discussion

Tobacco is one of the major target models for the development of transgenic plants [16,21,22]. The CRISPR-Cas9 method, an emerging gene-editing technology, remains a widely used tool for assessing whether desired mutations will occur and whether these mutations will be passed on to the next generation. Genetic tendencies have been reported in tomatoes, *Arabidopsis*, rice, and potatoes [12,13,23,24,25,26]. We systematically analyzed the mutation rates, types, and genetic trends in tobacco, in which editing of the *NbXylT1* and *NbFucT1* genes was induced by *Agrobacterium*-mediated CRISPR-Cas9.

Our gene-editing results revealed an average mutation rate of 23.7% (22.5% and 25% for *NbFucT1* and *NbXylT1* targets, respectively) (Table 1). In contrast, mutation efficiencies of approximately 80% have been obtained in *Arabidopsis*, rice, and tomato using the same CaMV35S promoter to drive Cas9 [12,25,27]. Although the rates obtained in the present study are notably lower than these values, they are slightly higher than the average efficiency of 20% reported in *N. tabacum* [28]. In this regard, it has been found that the GC content of the sgRNA targeting sequence in the CRISPR-Cas9 system and the ordered position of sgRNA sequences in the gRNA cassette for multiple targets can affect sgRNA efficacy [12,23,29,30]. Notably, although there was a two-fold difference in the GC content of the sgRNA used to target the two genes in the present study (30% for sgRNA *NbFucT1* and 60% for sgRNA *NbXylT1*), we observed little differences in the respective mutation rates (22% and 25%). However, assuming that sgRNA *NbFucT1* with low GC content is located in an advantageous position in the gRNA cassette, it is conceivable that the mutation rate dependent on GC content and position could be sufficiently altered.

Our analysis of mutation characteristics based on sequencing of the target region in selected T_0_ mutants revealed that the most common types of induced edits were deletions and insertions. Notably, insertions occurred primarily as single-base additions, whereas in the case of deletions, between 1 and 11 or more base pairs were excised from the target sites. With respect to the mutant genotype, we found that more than 83% of *NbFucT1* targets and 94% of *NbXylT1* targets were characterized by a homozygous or bi-allelic mutation (Table 2). In *Arabidopsis* [29] and rice [23], a number of different mutant genotypes have been reported, from homozygote to bi-allele, heterozygote, chimera, and wild type. The predominance of homozygous or bi-allelic mutations detected in the present study would, thus, indicate that the mutations attributable to the CRISPR-Cas9 system were generated during the early single stem cell stage. Given the absence of wild-type alleles, homozygous or bi-allelic mutations are expected to reduce the potential for the appearance of new mutations or chimeras in the next generation.

For several selected T_0_ transformants, we examined the manner in which the mutant genotype was transmitted to the T_1_ generation. This enabled us to establish that 81.8% of the T_1_ plants regenerated from homozygote mutant-type T_0_ plants inherited the same mutant form and that 77.4% of the T_1_ progeny regenerated from bi-allele mutant-type T_0_ individuals inherited the existing mutation at a level close to the expected segregation ratio (Table 3 and Table 4). Accordingly, this enabled us to confirm that a significant proportion of T_1_ lineage plants stably inherited the mutant trait from the previous generation, in accordance with Mendelian laws. Nevertheless, we detected smaller proportions of T_1_ progeny characterized by chimerism (approximately 10%) or unexpected additional mutations (approximately 10%) (Table 3 and Table 4), which are phenomena that have been reported in several previous studies [12,24,25,30]. To gain further insight into these variant mutation patterns, we screened selected individuals from the T_1_ generation lines for the presence of Cas9 (Appendix A). We reasoned that if Cas9 is removed by segregation, the target mutation can be stably transmitted to the next generation via the germline in accordance with Mendel’s laws in the absence of additional mutations in the progeny genotype. Conversely, if Cas9 is inherited from the previous generation, new mutations could potentially be induced by Cas9 in individuals with genotypes characterized by remaining wild-type alleles, such as heterozygous and chimeric types, which can lead to chimerism and unexpected segregation of target mutations [30]. However, in homozygous or bi-allelic individuals, even in the presence of Cas9, there is no editable wild-type target for CRISPR-Cas9; thus, no further mutation is likely to occur, and the target mutations can be stably inherited in the next generation [25]. In some T_1_ individuals regenerated from wild-type Ta75 and Ta139 T_0_ plants harboring an unmutated *NbFucT1* target region, we detected chimeric mutations generated by the activation of the delivered Cas9 (Table 3).

Our analyses revealed that some T_1_ plants were characterized by additional mutations, even in the absence of Cas9, whereas in some T_1_ plants regenerated from a T_0_ line that no longer carried wild-type alleles, we detected additional unexpected mutations in the presence of Cas9. The reason for unexpected additional mutations detected in representative genotypes and genetic flows can be considered from various aspects. First, it is possible that the position at which the CRISPR-Cas9 construct is inserted in the host genome affects the activity and rate of function of the CRISPR-Cas9 system. Consequently, even in plants characterized by a wild-type genotype at the beginning of the T_0_ generation, it is conceivable that a chimeric-type mutation may be generated due to the activation of Cas9 prior to the end of growth or in the subsequent T_1_ generation. A further point to consider with respect to mutation analysis is that the single leaf/cotyledon samples used for sequencing may not be representative of all genotypes in the plant being analyzed [22]. Consequently, we cannot exclude the possibility that the mutant type, which was presumed/diagnosed as the bi-allele type based on sequencing, is a chimera comprising a combination of homozygous and heterozygous genotypes in different cells [24].

Additionally, an off-target effect of Cas9 may occur, even in mutated plants. In a mutant that no longer has a wild-type allele due to sufficient mutation in the target site, if Cas9 is still present, the possibility of unexpected mutations due to off-target effects that allow homology mismatch should be considered. The CRISPR/Cas-based immune system has evolved while allowing a certain degree of homology mismatch in the target sequence to enhance defense efficiency but is more tolerant to higher mismatch rates [31,32]. Thus, the off-target effect is inversely related to the mismatch between on-target and potential off-target sequences [31]. The probability of off-target occurrence was reported to be 59% for one mismatch, 26% for two mismatches, and 3% for three mismatches [33]. In addition, a higher potential off-target effect may occur, depending on the mismatch position proximal to the PAM site, in which the further the mismatch is positioned from the PAM, the higher the off-target effect [31,32,33]. Based on precise gRNA design for consideration of out-of-frame scores, efficiency scores, and off-target mismatch counts, the off-target effect is negligible or less than 3% [31,33].

In the progeny of the evaluated bi-allele-type plants, unexpected additional mutations may occur regardless of the presence of Cas9, which may result in unequal inheritance. Given that the genotype of the T_1_ progeny is predicted based on that of the parent T_0_ plant, representative and accurate T_0_ genotyping is essential for minimizing the appearance of unexpected mutations in the T_1_ progeny and ensuring the generation of stable T_1_ regenerating lines. To ensure the selection of plants that have stable genetic flow, it is important to initially select a Cas9-free line in the T_1_ generation to eliminate further editing that may potentially recur in later generations. This is the ideal conclusion not only for tobacco but also for Cas9-induced mutants of other plants previously investigated [12,13,23].

In the present study, we traced the inheritance of mutations in 20 T_2_ generation individuals regenerated from the Ta161-1 plant, which had a homozygous mutant genotype and retained Cas9 (based on the analysis of gDNA extracted from whole plant samples, which were subjected to targeted deep sequencing) (Table 5). Accordingly, we detected no new mutations or aberrant segregation in any of the 20 T_2_ plants derived from this homozygous T_1_ plant and confirmed that Cas9 was isolated at a segregation ratio of 1:1, demonstrating that 50% of the progenies had Cas9-free status. In conclusion, the results of analyzing the genotypes of the T_0_, T_1_, and T_2_ generations of the Ta161 line support the hypothesis that mutations generated by the CRISPR-Cas9 method can be stably transmitted to the next generation when the progenies recover homozygous alleles.

Our findings will contribute to a better understanding of the behavior of Cas9 used in targeted mutagenesis in *N. benthamiana* and serve as a useful reference for developing strategies designed to ensure the production of stable homozygous mutant plants. Although we did not obtain tobacco mutants producing nontoxic N-glycan-type glycoproteins from glycosyltransferase enzymes (α-1,3-fucosyltransferase and β-1,2-xylosyltransferase), our results provide insights into the selection strategy of mutants without further segregation, regardless of the presence or absence of Cas9 protein.

## 4. Materials and Methods

### 4.1. Selection of sgRNA Target Sequences and Plasmid Construction

Among the five *NbFucT* and two *NbXylT* homologous genes, *NbFucT1* and *NbXylT1* were specifically targeted to detect any mutagenesis associated with glycosylation hindrance. Information on the sequences of *NbFucT1* (accession number: EF562630) and *NbXylT1* (accession number: EF562628) was obtained from data registered in GenBank (https://www.ncbi.nlm.nih.gov/genbank/, accessed on 1 May 2020). The target sequence of each gene was designed using CRISPOR software (http://crispor.tefor.net/, accessed on 1 May 2020). Through this program, sgRNA was selected by considering out-of-frame scores, efficiency scores, and off-target mismatch counts. In particular, we used at least three mismatched sgRNAs to reduce the rate of inducing off-target cleavage to below 3%, considering off-target effects, which is the most controversial part of the CRISPR-Cas9 method [33]. Two single guide RNAs (sgRNAs) were designed and linked to the *Arabidopsis* U6 promoter, which was assembled into a single pGenovo111 vector (Genovo Bio, Tianjin, China). A positive pNbFUT1-NbXYLT1 plasmid was selected and introduced into *Agrobacterium tumefaciens* strain EHA105 for subsequent transformation of tobacco plants.

### 4.2. Agrobacterium-Mediated Transformation of N. benthamiana Plants

To generate *Agrobacterium*-mediated transgenic tobacco, the first true leaves of 40-day-old tobacco (*N. benthamiana*) seedlings were cut and inoculated with a suspension (OD_600_ = 0.6–0.8) of *A. tumefaciens* harboring the CRISPR-Cas9 constructs. After incubation for 3 days at 25 °C in shoot induction medium containing 0.3% Gelrite in 200 µM acetosyringone (AS) supplemented with 1.0 mg/L BAP and 0.1 mg/L IAA, the seedlings were transferred to selective shoot initiation medium containing 150 mg/L Timentin and 20 µg/mL hygromycin for 2 weeks, after which they were transferred onto MS selective shoot elongation medium containing 0.3 mg/L BAP, 0.1 mg/L IAA, and 0.3% Gelrite. After reaching a height of approximately 3 cm, the seedlings were transferred to a selective rooting medium containing half-strength MS, 150 mg/L Timentin, and 15 mg/L hygromycin, and then incubated in a growth chamber under a 16 h light/8 h dark photoperiod, day and night temperatures of 20 °C to 24 °C, and relative humidity of 40% to 60%, respectively. After approximately 2 weeks, rooted regenerants were transferred to soil and grown to maturity in the growth chamber under the same conditions. For the selected lines, seeds were produced through self-pollination and used for inheritance analysis.

### 4.3. DNA Extraction and Identification of Induced Mutations

Total genomic DNA was extracted from 100 mg of leaf tissue using the DNeasy Plant Mini Kit (QIAGEN, Hilden, Germany). Using the extracted gDNA as a template, a fragment containing the target sites of *NbFucT1* and *NbXylT1* was amplified using gene-specific primer pairs (primer information is listed in Appendix A). PCR products were screened for mutations using the Sanger method or next-generation-sequencing-based targeted deep sequencing. When using the Sanger method, the PCR products were directly analyzed using internal sequencing primers (Appendix A) or inserted into a T&A cloning vector (RBC, Banqiao, Taiwan), with 5–10 clones sequenced for each sample. For targeted deep sequencing, target enrichment and library generation were performed using Phusion polymerase (New England Biolabs, Ipswich, MA, USA) and then analyzed using the Illumina MiSeq platform. Single-nucleotide polymorphisms and insertions/deletions in the target sequences identified from the sequencing data were analyzed using CRISPR RGEN tools (http://www.rgenome.net/cas-designer/, accessed on 1 August 2021) [34]. Based on the results of the mutagenesis analysis, zygosity types were classified as homozygous (when one type of mutation accounted for more than 98%), heterozygous (when one type of mutation was present in a proportion of 40–50%), bi-allelic (when two types of mutations were present in a proportion of 40–50% each), and chimeric (when two or more multiple mutations were mixed in various ratios) (5–30%).

### 4.4. Analysis of the Presence of Cas9

For all individuals of the T_1_ and T_2_ generation lines derived from plants that had been subjected to mutation analysis, the presence of Cas9 was determined based on gDNA PCR. The standard PCR amplification conditions were as follows: initial denaturation at 94 °C for 3 min; 35 cycles at 94 °C for 30 s, 55 °C for 30 s, and 72 °C for 30 s; and final extension at 72 °C for 7 min. The forward and reverse primers used were Cas9-F:5′-CCCACCATCTACCATCTGCG-3′ and Cas9-R:5′-ATGTCCTCGTTCTCCTCGTTGT-3′, respectively, and the expected product size was 1448 bp.

## Figures and Tables

**Figure 1 ijms-23-02450-f001:**
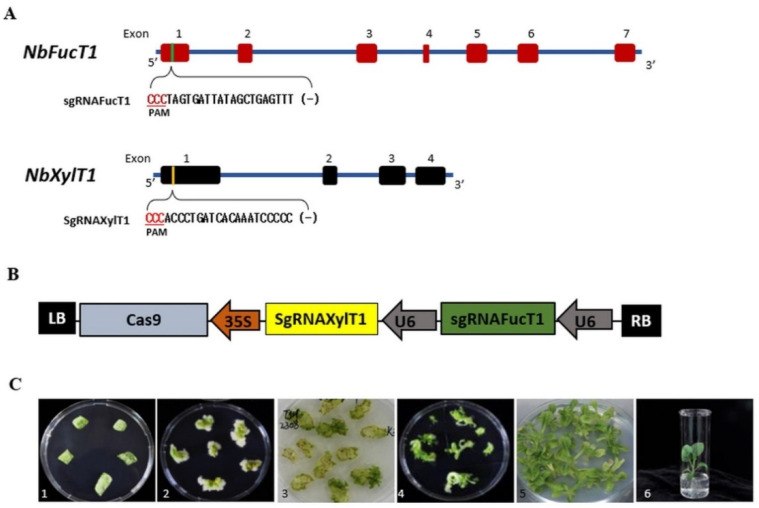
Transgenic tobacco plant production based on the application of CRISPR-Cas9-mediated gene editing. (**A**) Location of target sites in the *NbFucT1* (Accession number: EF562630) and *NbXylT1* (Accession Number: EF562628) genes and information on each gRNA sequence. Red and black boxes represent exons, and the sequence shown under exon 1 is that of gRNA. Nucleotides denoted in red are the PAM sequences. (**B**) Schematic diagram of a binary vector designed for simultaneous mutagenesis of the *NbFucTI* and *NbXylT1* genes using CRISPR-Cas9. (**C**) Photographs depicting the stages of transgenic tobacco plant production. 1. Infection and cocultivation; 2 and 3. Shoot induction; 4 and 5. Shoot elongation; 6. Rooting.

**Figure 2 ijms-23-02450-f002:**
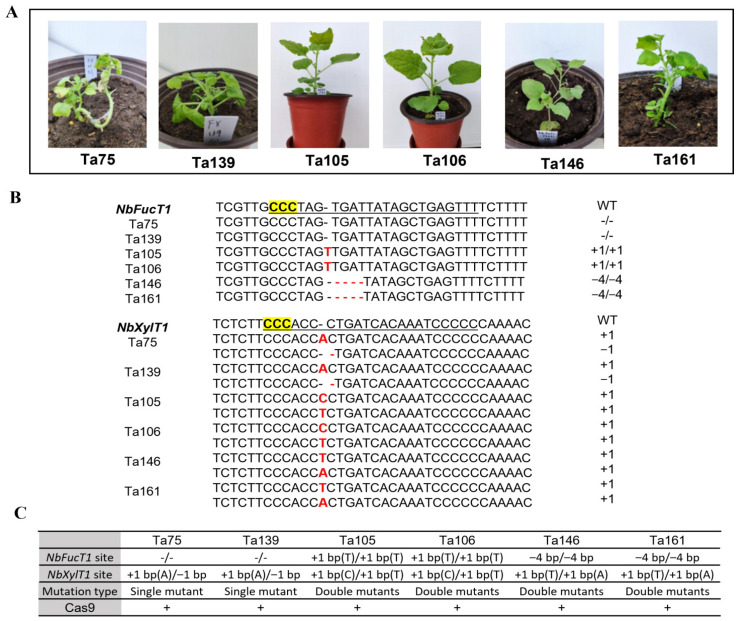
Selected examples of regenerated transgenic tobacco plants and their corresponding edited gene sequences. (**A**) Photographs of transgenic tobacco plants 2 weeks after transferring to soil. Plants were grown in pots and reached a height of 8.5 cm and a diameter of 9.5 cm at the top. Although some T_0_ plants showed shrunken appearances compared to the wild-type, in the case of T_1_ and T_2_ plants grown from seeds of the self-pollinated T_0_ and T_1_ plants, respectively, no phenotypic differences were found between the wild-type and mutated descendants. (**B**) Content and pattern of mutations in the targeted sequence. Nucleotides highlighted in yellow are PAM sequences, and underlined bases are gRNA sequences. The red letter indicates mutations in the target sequence. The + or − symbol in the target gene sequence represents base insertion or deletion compared to WT sequences, respectively. (**C**) Table of mutation information of transgenic plants. In the row of Cas9, a symbol + is for the presence of Cas9 in the mutants.

**Table 1 ijms-23-02450-t001:** Proportions of T_0_ plants with mutations in the target sequences.

Target Gene	No. of Plants Examined	No. of Mutated Regenerants			Mutation Rate by Locus		
		*NbFucT1*	*NbXylT1*	Both Loci	*NbFucT1*	*NbXylT1*	Both Loci
*NbFucT1*	248	56	62	51	22.5%	25%	20.56%
*NbXylT1*							

**Table 2 ijms-23-02450-t002:** Genotype and zygosity analysis of independent T_0_ transformants.

	Target Gene: *NbFucT1*				Target Gene: *NbXylT1*			
Mutant	Detected Mutations	Mutation Type	Total	Ratio	Detected Mutations	Mutation Type	Total	Ratio
Ta75	No mutation	Wild type	3	17.6%	+ 1bp/−1 bp	Bi-allele	16	94.%
Ta139	No mutation	Wild type			+1 bp/−1 bp	Bi-allele		
Ta158	No mutation	Wild type			+1 bp/+1 bp	Bi-allele		
Ta3	−9 bp/−9 bp	Homozygote	9	52.9%	−11 bp/+1 bp	Bi-allele		
Ta12	−9 bp/−9 bp	Homozygote			−11 bp/+1 bp	Bi-allele		
Ta18	−9 bp/−9 bp	Homozygote			−11 bp/+1 bp	Bi-allele		
Ta27	−9 bp/−9 bp	Homozygote			−11 bp/+1 bp	Bi-allele		
Ta105	+1 bp/+1 bp	Homozygote			+1 bp/+1 bp	Bi-allele		
Ta106	+1 bp/+1 bp	Homozygote			+1 bp/+1 bp	Bi-allele		
Ta146	−4 bp/−4 bp	Homozygote			+1 bp/+1 bp	Bi-allele		
Ta161	−4 bp/−4 bp	Homozygote			+1 bp/+1 bp	Bi-allele		
Ta122	−113 bp/−113 bp	Homozygote			+1 bp/+1 bp	Bi-allele		
Ta117	+1 bp/−9 bp	Bi-allele	5	29.4%	+1 bp/−9 bp	Bi-allele		
Ta133	+1 bp/−9 bp	Bi-allele			+1 bp/−9 bp	Bi-allele		
Ta237	+1 bp/−4 bp	Bi-allele			+1 bp/−9 bp	Bi-allele		
Ta248	+1 bp/+1 bp	Bi-allele			+1 bp/+1 bp	Bi-allele		
Ta77	+1 bp/−4 bp	Bi-allele			No Mutation	Wild type	1	5.8%

**Table 3 ijms-23-02450-t003:** Genetic analysis of CRISPR-Cas9 induced mutations of *NbFucT1* in the T_0_ generation and their transmission to the T_1_ generation.

Target Gene: *NbFucT1*						
T_0_ Generation		T_1_ Generation				
Plant ID	Zygosity (Detected Variation)	Plant ID	Zygosity	Detected Variation	No. of Plants	Ratio
Ta75	Wild type (no mutation)		Bi-allele		0	0.0%
			Homozygote		0	0.0%
		75-2, 75-12, **75-14**, 75-15 75-16, **75-18**	Chimera	+1 bp/−6 bp/−5 bp/−4 bp * WT/+1 bp/−6 bp/−5 bp/−4 bp *	4 2	36.4% 18.2%
		**75-3**, **75-4**, **75-11**, 75-13, **75-17**	Wild type	No mutation	5	45.5%
Ta139	Wild type (no mutation)		Bi-allele		0	0.0%
			Homozygote		0	0.0%
		139-2, 139-16 139-1	Chimera	+1 bp/−6 bp/WT/−5 bp/−4 bp * WT/+1 bp/−6 bp/−5 bp/−4 bp *	2 1	25.0% 12.5%
		**139-11**, **139-12**, **139-13**, **139-14**, **139-15**	Wild type	No mutation	5	62.5%
Ta105	Homozygote (+1 bp(T)/+1 bp(T))	105-4	Bi-allele	+1 bp(G)/ −1 bp *	1	10.0%
		**105-2**, **105-11**, **105-12**, **105-14**, **105-15**, 105-16, 105-17, **105-18**	Homozygote	+1 bp(T)/+1 bp(T)	8	80.0%
		105-3		+1 bp(G)/+1 bp(G) *	1	10.0%
			Chimera		0	0.0%
			Wild type	No mutation	0	0.0%
Ta106	Homozygote (+1 bp(T)/+1 bp(T))		Bi-allele		0	0.0%
		**106-2**, **106-12**, **106-13**, **106-14**, **106-15**, **106-17**, **106-18**	Homozygote	+ 1bp(T)/+1 bp(T)	7	87.5%
		106-1	Chimera	−5 bp/−2 bp/−4 bp/+1 bp *	1	12.5%
			Wild type	No mutation	0	0.0%
Ta146	Homozygote (−4 bp/−4bp)	146-16	Bi-allele	−4 bp/+1 bp(T) *	1	10.0%
		146-1, 146-2, 146-11, 146-12, 146-13, 146-17, 146-18	Homozygote	−4 bp/−4 bp	7	70.0%
		146-14 **146-15**	Chimera	+1 bp(T)/−6 bp/−5 bp/−4 bp * −3 bp/+1 bp(T)/−6 bp/−5 bp *	1 1	10.0% 10.0%
			Wild type	No mutation	0	0.0%
Ta161	Homozygote (−4 bp/−4 bp)		Bi-allele		0	0.0%
		161-1, 161-2, **161-11**, 161-12, 161-13, **161-14**, 161-15, 161-16, **161-18**	Homozygote	−4 bp/−4 bp	9	90.0%
		**161-17**	Chimera	+1 bp(T)/−6 bp/−5 bp/−4 bp *	1	10.0%
			Wild type	No mutation	0	0.0%

The plant IDs denoted in bold type indicate Cas9-free plants. * Unexpected mutations.

**Table 4 ijms-23-02450-t004:** Genetic analysis of CRISPR-Cas9-induced mutations of *NbXylT1* in the T_0_ generation and their transmission to the T_1_ generation.

Target Gene: *NbXylT1*						
T_0_ Generation		T_1_ Generation				
Plant ID	Zygosity (Detected Variation)	Plant ID	Zygosity	Detected Variation	No. of Plants	Ratio
Ta75	Bi-allele (+1 bp(A)/−1 bp)	**75-11**, **75-14**, 75-15, **75-17** 75-16	Bi-allele	+1 bp(A)/−1 bp +1 bp(T)/−1 bp *	4 1	36.4% 9.1%
		75-2, 75-12 **75-3**, **75-4**, 75-13	Homozygote	−1 bp/−1 bp +1 bp(A)/+1 bp(A)	2 3	18.2% 27.3%
		**75-18**	Chimera	+1 bp(T)/−1 bp/+1 bp(A) *	1	9.1%
			wild type	no mutation	0	0.0%
Ta139	Bi-allele (+1 bp(A)/−1 bp)	139-1, **139-11**, **139-12** **139-15**	Bi-allele	+1 bp(A)/ −1 bp +1 bp(A)/+1 bp(T) *	3 1	37.5% 12.5%
		**139-13**, **139-14**, 139-16	Homozygote	+1 bp(A)/+1 bp(A)	3	37.5%
		139-2	Chimera	+1 bp(A)/+1 bp(T)/ −1 bp *	1	12.5%
			Wild type	WT	0	0.0%
Ta105	Bi-allele (+1 bp(T)/+1 bp(C))	**105-2**, **105-11**, **105-14**, **105-15**, 105-17	Bi-allele	+1 bp(T)/+1 bp(C)	5	50.0%
		**105-12** 105-16 105-3, 105-4	Homozygote	+1 bp(T)/+1 bp(T) +1 bp(C)/+1 bp(C) +1 bp(A)/+1 bp(A) *	1 1 2	10.0% 10.0% 20.0%
		**105-18**	Chimera	+1 bp(T)/+1 bp(A)/−1 bp *	1	10.0%
			Wild type	WT	0	0.0%
Ta106	Bi-allele (+1 bp(T)/+1 bp(C))	**106-1**, **106-2**, **106-13**	Bi-allele	+1 bp(T)/+1 bp(C)	3	37.5%
		**106-14**, **106-15**, **106-17** **106-12**	Homozygote	+1 bp(T)/+1 bp(T) +1 bp(C)/+1 bp(C)	3 1	37.5% 12.5%
		**106-18**	Chimera	+1 bp(A)/+1 bp(T)/+1 bp(C) *	1	12.5%
			Wild type	WT	0	0.0%
Ta146	Bi-allele (+1 bp(A)/+1 bp(T))	146-1, 146-11, 146-12, 146-16, 146-17 **146-15**	Bi-allele	+1 bp(A)/+1 bp(T) −1 bp/+1 bp(T) *	5 1	50.0% 10.0%
		146-13	Homozygote	+1 bp(A)/+1 bp(A)	1	10.0%
		146-2, 146-14, 146-18	Chimera	+1 bp(A)/+1 bp(T)/−1 bp *	3	30.0%
			Wild type	WT	0	0.0%
Ta161	Bi-allele (+1 bp(A)/+1 bp(T))	**161-11**, 161-13, 161-15	Bi-allele	+1 bp(A)/+1 bp(T)	3	30.0%
		161-1, 161-2, 161-12, 161-16, **161-17** **161-14**	Homozygote	+1 bp(T)/+1 bp(T) +1 bp(A)/+1 bp(A)	5 1	50.0% 10.0%
		**161-18**	Chimera	+1 bp(A)/+1 bp(T)/−1 bp *	1	10.0%
			Wild type	WT	0	0.0%

The plant IDs denoted in bold type indicate Cas9-free plants. * Unexpected mutations.

**Table 5 ijms-23-02450-t005:** Analysis of the transmission of homozygous mutant Ta161 for *NbXylT1* and *NbFucT1* loci from the T_1_ to T_2_ generations.

Generation			*NbFucT1*		*NbXylT1*	
	Plant ID	cas9	Mutation Detected	Mutation Type	Mutation Detected	Mutation Type
T_0_ plant	Ta161	+	+1 bp(T)/+1 bp (A)	Bi-allelic	−4 bp/−4 bp	Homozygous
T_1_ plant	Ta161-1	+	+1 bp(T)/+1 bp (T)	Homozygous	−4 bp/−4 bp	Homozygous
T_2_ plant	Ta161-1-1	+	+1 bp(T)/+1 bp (T)	Homozygous	−4 bp/−4 bp	Homozygous
	Ta161-1-2	−	+1 bp(T)/+1 bp (T)	Homozygous	−4 bp/−4 bp	Homozygous
	Ta161-1-3	−	+1 bp(T)/+1 bp (T)	Homozygous	−4 bp/−4 bp	Homozygous
	Ta161-1-4	+	+1 bp(T)/+1 bp (T)	Homozygous	−4 bp/−4 bp	Homozygous
	Ta161-1-5	+	+1 bp(T)/+1 bp (T)	Homozygous	−4 bp/−4 bp	Homozygous
	Ta161-1-6	−	+1 bp(T)/+1 bp (T)	Homozygous	−4 bp/−4 bp	Homozygous
	Ta161-1-7	−	+1 bp(T)/+1 bp (T)	Homozygous	−4 bp/−4 bp	Homozygous
	Ta161-1-8	+	+1 bp(T)/+1 bp (T)	Homozygous	−4 bp/−4 bp	Homozygous
	Ta161-1-9	−	+1 bp(T)/+1 bp (T)	Homozygous	−4 bp/−4 bp	Homozygous
	Ta161-1-10	+	+1 bp(T)/+1 bp (T)	Homozygous	−4 bp/−4 bp	Homozygous
	Ta161-1-11	+	+1 bp(T)/+1 bp (T)	Homozygous	−4 bp/−4 bp	Homozygous
	Ta161-1-12	+	+1 bp(T)/+1 bp (T)	Homozygous	−4 bp/−4 bp	Homozygous
	Ta161-1-13	+	+1 bp(T)/+1 bp (T)	Homozygous	−4 bp/−4 bp	Homozygous
	Ta161-1-14	−	+1 bp(T)/+1 bp (T)	Homozygous	−4 bp/−4 bp	Homozygous
	Ta161-1-15	−	+1 bp(T)/+1 bp (T)	Homozygous	−4 bp/−4 bp	Homozygous
	Ta161-1-16	+	+1 bp(T)/+1 bp (T)	Homozygous	−4 bp/−4 bp	Homozygous
	Ta161-1-17	−	+1 bp(T)/+1 bp (T)	Homozygous	−4 bp/−4 bp	Homozygous
	Ta161-1-18	−	+1 bp(T)/+1 bp (T)	Homozygous	−4 bp/−4 bp	Homozygous
	Ta161-1-19	+	+1 bp(T)/+1 bp (T)	Homozygous	−4 bp/−4 bp	Homozygous
	Ta161-1-20	−	+1 bp(T)/+1 bp (T)	Homozygous	−4 bp/−4 bp	Homozygous

+ or − in the Cas9 column indicates the presence or absence of Cas9 in the plant, respectively.

## Data Availability

Data is contained within the article or Appendix A.

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
