# Peer review of "Genetic Dissection of CRISPR-Cas9 Mediated Inheritance of Independently Targeted Alleles in Tobacco α-1,3-Fucosyltransferase 1 and β-1,2-Xylosyltransferase 1 Loci"

_ijms, 2022, doi:10.3390/ijms23052450_

Round 1
Reviewer 1 Report
The authors have fully addressed my concerns. I only have some very minor comments left for the manuscript:
Line 18 NbFucT1 and NbXylT1 is first time shown in paper, thus full name should be given.
Line 25 If I understand correctly '+1bp insertion or -4bp deletion' should be '+1bp insertion and -4bp deletion'. +1 bp insertion in NbFucT1 and -4bp deletion NbXylT1?
Line 119 NbFucT1 or NbXylT1 gene should be italic.
Reviewer 2 Report
The creation of improved varieties of cultivated plants with increased productivity, adapted to various adverse conditions, is a priority in agricultural sciences. There are several technologies, but they all have their own advantages and disadvantages. There are such in the new method - CRISPR-Cas9. One of them is whether point mutations in a gene will lead to the desired changes and whether these mutations will be passed on to the next generation. The authors tried to answer the last question, taking the transferase genes as a basis. A lot of work has been done, interesting findings have been obtained that will help researchers in the future when using this method. There is no doubt about the obtained results and conclusions.
There are a number of comments.
1. What changes occur in tobacco after mutations? Photos of plants are given, however, there is no detailed description for them, neither in the methods section, nor in the discussion section.
2. in fig. 1 and 2 are photos of plants, but there is no data on either the age of the plants or the conditions for their cultivation.
3. glycosyltransferases are enzymes that use glycosides as donors for transfer to acceptors. Xylose and fucose are designations for monosaccharides, not glycosides. According to the nomenclature, it is necessary to designate β-1,2-xycoside and α-1,3-fucoside. And do not repeat the mistake of the authors in reference 19 (lines-109, 351 add a glycosidic bond at the fucoside, 416 should be changed to α and β ).
4. Tables 3 and 4 relationship what to what?
5. line 207 - not clear 54% of mutations occur in the wild type in the next generation? Is it really true?
Table 3 indicates that no mutation occurs in the wild type.
6. Were any quantitative measurements of N-glucan carried out and how did point mutations of transferase genes affect n-glucan?
7. The choice of target is very important for the CRISPR-Cas9 method. Is there a homology of the chosen target with other sequences in tobacco and what is their %?
8. reference 31, indicate the year in bold and reference 32 change color.
Author Response
Please see the attachment.

This manuscript is a resubmission of an earlier submission. The following is a list of the peer review reports and author responses from that submission.
Round 1
Reviewer 1 Report
The paper submitted by Song etc. described a comprehensive study of the inheritance of tobacco plants edited with CRISPR/Cas9 (NbFucT1 and NbXylT1 genes). With the development of gene editing technology, this study will be of great interest to broad range of readers, while also be ignored frequently. Major flaw of this study is that the author didn't give detailed background information, include expression level and copy number of Cas9 proteins in different transgenic lines, which makes this paper hard to get very conclusive conclusions. For example, is it possible that some T1 plants still have Cas9, while some not. And these have Cas9 plants, gene editing is still happening and results in unexpected segregation ratio? The copy number or expression level of Cas9 proteins may also led to this phenomenon, while no data to exclude or confirm these possibilities.
Minor points:
1 Title should be "CRISPR/Cas9"
2 L40: foreign genes could be exogeneous genes
3 Ll163: "Inheritance of T0 variant target mutations by T1 plants" could be "Inheritance of T0 variant target mutations evaluated in T1 plants".
4 L166: are these T0 plants Cas9 free? How many copies of Cas9 in these lines?
L191: number of samples (n) should be listed.
5 L305-309: is it possible that some of Cas9 sites have been silent by RNAi or other mechanisms, while some may not and continuously editing on/off-target sites? I think the authors should check whether Cas9 proteins in lines with different inheritance is still activate or silent. Furthermore, the expression of CAS9 is driven by 35S promoter, which may have different expression level in transgenic lines and result in different pattern of inheritances as observed in this study.
Reviewer 2 Report
In this manuscript Song et al perform CRISPR-Cas9 genome editing in Tobacco and track the inheritance of these edits until the F2 generation. I have only minor comments:
- CRISPR/Cas9 is not the correct nomenclature. Use CRISPR-Cas9 throughout including the title
- The authors should reconsider their claim of assessing specificity. To be able to confidently assess specificity, one would need Whole genome sequencing
Round 2
Reviewer 1 Report
All my concerns have been addressed appropriately.
